# RANTES 59029A/G Polymorphisms Associated with Diabetic Compilations in Korean Patients with Type 2 Diabetes for over 15 Years

**DOI:** 10.3390/genes12091445

**Published:** 2021-09-19

**Authors:** Dong-Hwa Lee, Eu-Jeong Ku, Tae-Keun Oh, Hyun-Jeong Jeon

**Affiliations:** Department of Internal Medicine, Chungbuk National University Hospital, Cheongju 28644, Korea; roroko@hanmail.net (D.-H.L.); eujeong.ku@gmail.com (E.-J.K.); tgohkjs@chungbuk.ac.kr (T.-K.O.)

**Keywords:** CCL5, RANTES, single nucleotide polymorphism, type 2 diabetes mellitus, diabetic complications

## Abstract

Background: Polymorphisms in the RANTES gene are known to be associated with several diseases related to insulin resistance. In this study, we investigated the association between RANTES 59029A/G polymorphisms and the prevalence of diabetic complications relative to obesity in Korean patients who had type 2 diabetes (T2D) for over 15 years. Methods: A single-center, retrospective case-control study was performed. We included 271 patients with a duration of diabetes greater than 15 years. Polymerase chain reaction-restriction fragment length polymorphism was used to analyze RANTES polymorphisms, identifying genotypes as GG, AG, or AA. Obesity was defined using the body mass index with a cutoff value of 25 kg/m^2^. Both microvascular (retinopathy and nephropathy) and macrovascular (coronary artery disease and cerebrovascular disease) complications were evaluated. Results: The duration of T2D and hemoglobin A1c values at enrollment were 24.4 ± 5.0 years and 7.8 ± 1.6%, respectively, in the non-obese group, and 25.4 ± 6.1 years and 7.7 ± 1.7%, respectively, in the obese group. The prevalence of microvascular complications was significantly higher in the obese group compared with that in the non-obese group (83.5% vs. 72.0%, *p* = 0.039). Compared to the non-obese group, the obese group showed a higher proportion of the patients with AA or AG genotypes (64.3% vs. 84.5%, *p* = 0.001). Conclusions: The A allele of the RANTES gene is associated with obesity and may affect diabetic microvascular complications in patients with T2D for over 15 years.

## 1. Introduction

Currently, there is a tremendous number of patients that suffer from diabetes. The global prevalence of diabetes in adults in 2019 was reported to be 463 million, 9.3% of the adult population, and this is expected to increase to 10.2% (578 million) by 2030 and 10.9% (700 million) by 2045 [1]. An increase in the prevalence is observed in Korea as well. The prevalence of diabetes in Korea in 2017 was reported to be about 10% of the adult population, which was slightly higher than the global prevalence [2]. In line with the increasing prevalence, there is also a concern for increasing diabetic complications as it may affect morbidity and mortality of diabetic patients. Diabetes is among the top 10 causes of death in adults, with it estimated to have caused four million deaths globally in 2017 [3]. In Korea, diabetes was the sixth leading cause of death both in men and women [4]. Therefore, preventing the development and progression of complications is an important goal in diabetic management.

Several traditional risk factors are well-known for their association with diabetes prevalence, sedentary lifestyle, dyslipidemia, and hypertension. Among them, obesity is an important risk factor for metabolic disorders, including type 2 diabetes (T2D). Obesity is thought to be attributed to interactions between environmental and genetic factors. According to a previous report, genetic factors are responsible for approximately 40–70% of the etiology of obesity [5]. Previous studies have reported that polymorphisms in several genes are associated with obesity and diabetic complications in patients with T2D [6,7,8,9].

Previous studies have also shown that obesity and inflammation in adipose tissue are closely related to insulin resistance [10,11].

The protein regulated on activation, normal T-cell expressed and secreted (RANTES), which is also known as CC chemokine ligand 5 (CCL5), recruits and activates different subtypes of leukocytes, such as T cells, monocytes, basophils, eosinophils, and mast cells [12]. RANTES is expressed in adipose tissue, and therefore, may be involved in the development of obesity through adipose tissue inflammation [13]. Previous studies have shown that polymorphism in the RANTES gene and increased serum levels of RANTES are associated with the incidence of T2D, metabolic syndrome, and coronary artery disease, which are known to be induced by insulin resistance [14,15,16]. Another study involving patients with T2D demonstrated that RANTES polymorphism is associated with diabetic nephropathy [17].

Obesity and the long-term duration of T2D are both risk factors for developing diabetes complications. However, studies on the effects of RANTES polymorphisms on patients with a long-term duration of T2D, who are prone to complications, have been lacking. Furthermore, there were no studies evaluating the association among polymorphisms of the RANTES gene, obesity, and T2D. In the current study, we investigated whether RANTES polymorphism affected the prevalence of diabetic complications and its association with obesity in Korean patients that have had T2D for more than 15 years.

## 2. Materials and Methods

### 2.1. Study Population and Design

This was a single-center, retrospective case-control study. Korean patients with T2D were recruited from the endocrinology outpatient clinic of Chungbuk National University (Cheongju, South Korea). The selected patients had a duration of diabetes greater than 15 years. To determine the duration of T2D, the year of diagnosis was based on the patient’s memory. A total of 271 patients were included in the analysis. The classification of obesity was performed according to the body mass index (BMI) of the patient. BMI (kg/m^2^) was calculated as the body weight (kg) of the patient divided by the square of the patient’s height (m^2^). The patients were categorized into two groups, non-obese and obese, using a BMI of 25 kg/m^2^ as a cutoff value, according to Asian-Pacific guidelines [18]. Among the 271 enrolled patients, 103 were assigned to the obese group and 168 were assigned to the non-obese group. All participants provided written informed consent. The study was approved by the Institutional Review Board of the Chungbuk National University Hospital (IRB No.2017-10-009) and conducted according to the guidelines of the Declaration of Helsinki.

### 2.2. Evaluation of Diabetic Complications

We evaluated both microvascular complications, including retinopathy and nephropathy, and macrovascular complications, such as coronary artery disease (CAD) and cerebrovascular disease (CVD). Due to the study design being retrospective, most of the data was collected based on medical records and information obtained from the patients during history taking. Nephropathy was determined according to the estimated glomerular filtration rate (eGFR) using a cutoff value of 60 mL/min/1.73 m^2^. CAD was evaluated based on medical records and the results of coronary angiographic data. The presence of CAD was defined in patients who had a history of stable angina, unstable angina, myocardial infarction, or intervention for coronary artery disease.

### 2.3. Genotyping of RANTES 59029A/G

Genomic DNA was extracted from peripheral blood cells using a DNA Extraction Kit (Qiagen, Valencia, CA, USA). RANTES 59029A/G genotyping was performed using polymerase chain reaction (PCR)–restriction fragment length polymorphism analysis. Sequences of the RANTES-specific primers used for the analysis of the RANTES 59029A/G genotype were 5′-CCC GTG AGC CCA TAG TTA AAA CTC-3′ for the forward primer and 5′-TCA CAG GGC TTT TCA ACA GTA AGG-3′ for the reverse primer. Reactions were cycled using the following parameters: initial denaturation at 95 °C for 5 min, followed by 36 cycles each of 94 °C for 30 s, 59 °C for 30 s, and 72 °C for 50 s, and a final extension step of 72 °C for 10 min. The PCR-amplified products were digested with Bsp12861 restriction enzyme (Amersham Pharmacia Biotech, Amersham, UK). Complete digestion of the 268 bp PCR product into 138 and 130 bp fragments (homozygous cut) was interpreted as a GG genotype. In contrast, the AA genotype generated PCR-amplified products that lacked a Bsp12861 recognition sequence, making them indigestible and resulting in intact 268 bp PCR fragments (No-cut).

### 2.4. Statistical Analysis

The probability of Hardy–Weinberg equilibrium was tested using the chi-squared test. The data were expressed as the mean ± standard deviation (SD) or as percentages for the categorical variables. The baseline characteristics were compared using Student’s *t*-test for continuous variables and the chi-squared test for categorical parameters. All statistical analyses were performed using SPSS for Windows software, version 22.0 (IBM Corp., Armonk, NY, USA). Statistical significance was set at *p* < 0.05.

## 3. Results

### 3.1. Characteristics of the Study Population

The baseline demographic and clinical characteristics of patients according to BMI (non-obese vs. obese) are presented in Table 1. The mean duration of T2D was 24.4 ± 5.0 years in the non-obese group and 25.4 ± 6.1 years in the obese group (*p* = 0.163). The mean BMI was 22.2 ± 2.3 kg/m^2^ in the non-obese group and 26.7 ± 1.8 kg/m^2^ in the obese group (*p* = 0.001). Patients in the obese group were older on average than those in the non-obese group (62.1 ± 9.3 years vs. 59.2 ± 10.6 years, respectively; *p* = 0.025). The proportion of women was higher in the obese group compared to that in the non-obese group (64.1% vs. 44.0%, respectively; *p* = 0.001). The hemoglobin A1c (HbA1c) values at enrollment were not significantly different between the two groups (7.8 ± 1.6% in the non-obese group and 7.7 ± 1.7% in the obese group; *p* = 0.744). No statistically significant differences were observed between the two groups regarding other laboratory findings, including lipid parameters, liver enzymes, or kidney function.

### 3.2. Diabetic Complications Relative to Patient BMI

The prevalence of diabetic complications relative to BMI (non-obese vs. obese) is shown in Table 2. The proportion of patients with retinopathy was similar in the two groups (75.6% in the non-obese group and 79.6% in the obese group; *p* = 0.207). The prevalence of nephropathy was also similar at 50.3% in the non-obese group and 56.3% in the obese group (*p* = 0.320). However, the total number of patients who had diabetic microvascular complications was significantly higher in the obese group compared to that in the non-obese group (83.5% vs. 72.0%, respectively; *p* = 0.039). There were no statistically significant differences between the two groups regarding macrovascular complications, including the type (CAD and CVD) or prevalence of macrovascular complications.

### 3.3. RANTES Genotype Relative to Patient BMI

The distribution of RANTES genotypes was as follows: AA genotype, 25.8% (*n* = 70); AG genotype, 46.1% (*n* = 125); and GG genotypes, 28.0% (*n* = 76). The genotypic distribution of RANTES followed the Hardy–Weinberg equilibrium principle. Meanwhile, the frequency of RANTES genotypes exhibited different patterns relative to patient BMI (Table 3). Compared to the non-obese group, the obese group showed a higher proportion of patients with either AA or AG genotypes (64.3% vs. 84.5%, respectively; *p* = 0.001) (Figure 1A). According to allele frequency analysis, patients in the obese group had a higher frequency of A allele than G allele, whereas patients in the non-obese group had a higher frequency of G allele than A allele (*p* = 0.001) (Figure 1B).

## 4. Discussion

The current case-control retrospective study aimed to investigate the association between RANTES 59029A/G polymorphisms and obesity and the impact of the polymorphisms on diabetic complications in patients with long-term T2D (duration > 15 years). Obese patients with T2D exhibited a higher prevalence of diabetic microvascular complications. Moreover, patients with the AA or AG genotype of the RANTES gene, in other words with an A allele, were observed more frequently in the obese group compared to that in the non-obese group. This suggested that obesity was associated with the development of diabetic complications, and the RANTES polymorphism may have been a contributing factor in this process.

RANTES is known to be a potent chemoattractant for monocytes, lymphocytes, eosinophils, and basophils, contributing to the inflammatory process [19]. Previous studies have demonstrated that RANTES is associated with several diseases related to chronic inflammatory responses, such as atherosclerosis, rheumatoid arthritis, liver disease, and asthma [20,21,22,23]. Obesity is a typical chronic inflammatory-related disease induced by the infiltration of inflammatory cells into the adipose tissue. RANTES has been shown to have a close relationship with obesity, as well as with phenotypes associated with obesity [24,25,26,27]. The present study showed that different patterns of RANTES polymorphism were observed in obese patients with long-term T2D compared to those in non-obese patients, which aligned with results from the previous studies. We found that the A allele of RANTES 59029A/G was more frequently present in obese patients. Although obesity is a multifactorial disease, we believe that genetic factors may contribute to its development, as suggested by our current results.

Recent studies demonstrated that polymorphisms in the RANTES gene are associated with the incident of T2D [28,29]. According to one of the studies, the AA genotype is more frequently observed in patients with T2D compared to that in control patients [28]. Moreover, a significant correlation between RANTES polymorphisms and atherosclerosis has been confirmed [30,31]. Considering that atherosclerosis is an important pathophysiology associated with the development and progression of diabetic complications, RANTES polymorphism may to some degree affect diabetic complications. There have been a few studies conducted to investigate diabetic complications in patients with T2D that have focused on diabetic nephropathy. One study showed that RANTES is associated with diabetic nephropathy in patients with T2D based on the assessment of albuminuria and serum creatinine [32]. In a 10-year longitudinal study published by the same group, they confirmed RANTES 59029G/A polymorphisms exhibit an independent positive correlation with the onset or progression of nephropathy [17]. In our current study, a significant association was observed between obesity and diabetic microvascular complications, including nephropathy and retinopathy, but not nephropathy alone. However, there was no significant association between obesity and macrovascular complications. The inconsistent results compared to those of previous studies may be attributed to the population in our study. We performed our study using patients with long-term T2D, which made them vulnerable to diabetic complications and other diseases. Moreover, our results suggested that RANTES polymorphisms affected the development of obesity, which then influenced diabetic complications. Combining our current results with those of previous studies suggests that RANTES polymorphisms may contribute to diabetic complications at various stages of development and progression of the complications.

A strength of our current study was that we performed a comprehensive evaluation of diabetic complications, including both microvascular and macrovascular complications. However, the results of our study also had limitations, including it being a retrospective study with a relatively small number of subjects. Furthermore, due to the retrospective nature of the study, we were not able to determine serum levels of RANTES in the patients and thus could not determine whether genetic polymorphisms of RANTES affected the serum levels. The lack of a control group of the population without T2D is also a limitation of this study.

In conclusion, the presence of RANTES AA or AG genotypes was associated with obesity in Korean patients with long-term T2D of more than 15 years. Furthermore, obesity in these patients demonstrated a significant correlation with the prevalence of diabetic microvascular complications. Our results support that the presence of AA or AG genotypes in the RANTES gene may be involved as risk factors for diabetic complications in Korean patients that have T2D for longer durations. A prospective study to confirm the present results and basic research to clarify the underlining mechanism are needed.

## Figures and Tables

**Figure 1 genes-12-01445-f001:**
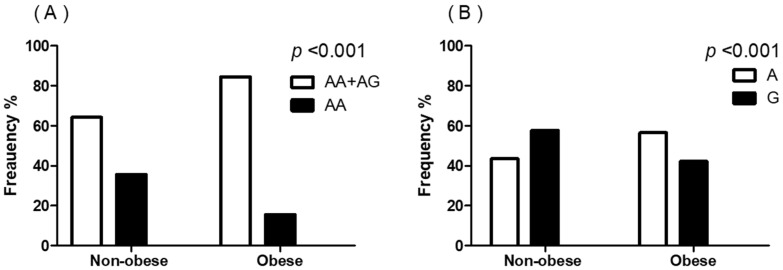
Frequency of RANTES genotypes and alleles according to the presence of obesity. (**A**) Distribution of genotypes of RANTES gene. (**B**) Frequency of alleles of RANTES gene. The *p*-values were calculated using the chi-squared test.

**Table 1 genes-12-01445-t001:** Baseline characteristics of patients relative to BMI.

	Non-Obese (*n* = 168)	Obese (*n* = 103)	*p*-Value
Age (years)	59.2 ± 10.6	62.1 ± 9.3	0.025
Sex (F/M)	74/94	66/37	0.001
Duration of DM (years)	24.4 ± 5.0	25.4 ± 6.1	0.163
BMI (kg/m^2^)	22.2 ± 2.3	26.7 ± 1.8	0.001
SBP (mmHg)	138.9 ± 16.6	137.8 ± 15.1	0.611
DBP (mmHg)	85.8 ± 10.8	86.0 ± 9.6	0.882
FPG (mg/dl)	146.2 ± 55.5	137.0 ± 45.5	0.215
PP2 (mg/dl)	248.3 ± 96.7	225.1 ± 89.2	0.093
HbA1c (%)	7.8 ± 1.6	7.7 ± 1.7	0.744
C-peptide (ng/mL)	4.20 ± 1.78	3.04 ± 1.83	0.618
Total cholesterol (mg/dL)	179.4 ± 46.0	189.9 ± 55.5	0.145
Triglycerides (mg/dL)	153.9 ± 81.5	191.2 ± 99.1	0.056
HDL-cholesterol (mg/dL)	44.7 ± 10.9	44.7 ± 12.0	0.996
LDL-cholesterol (mg/dL)	103.8 ± 32.6	108.7 ± 43.7	0.384
AST (IU/L)	38.2 ± 10.4	39.1 ± 15.4	0.901
ALT (IU/L)	39.2 ± 11.4	38.2 ± 11.4	0.801
BUN (mg/dL)	25.4 ± 21.3	26.3 ± 19.9	0.302
Creatinine (mg/dL)	2.32 ± 2.85	2.41 ± 2.68	0.803

Data are expressed as the mean ± SD. DM, diabetes mellitus; BMI, body mass index; SBP, systolic blood pressure; DBP, diastolic blood pressure; FPG, fasting plasma glucose; PP2, postprandial 2-h glucose; HbA1c, hemoglobin A1c HDL, high-density lipoprotein; LDL, low-density lipoprotein; AST, aspartate aminotransferase; ALT, alanine aminotransferase; BUN, blood urea nitrogen. The *p*-values were calculated using Student’s *t*-test for continuous data and the chi-squared test for categorical data.

**Table 2 genes-12-01445-t002:** Comparison of diabetic complications relative to BMI.

Complications, N (%)	Non-Obese (*n* = 168)	Obese (*n* = 103)	*p*-Value
Microvascular complications	121 (72.0)	86 (83.5)	0.039
Retinopathy	94 (75.6)	70 (79.6)	0.207
Nephropathy	45 (50.3)	58 (56.3)	0.320
Macrovascular complications	65 (38.7)	41 (39.8)	0.898
Coronary artery disease	48 (58.6)	33 (32.1)	0.682
Cerebrovascular disease	27 (16.7)	16 (15.5)	0.988

Data are expressed as number (%). The *p*-values were calculated using the chi-squared test.

**Table 3 genes-12-01445-t003:** Distribution of RANTES 59029A/G polymorphisms.

Variables, N (%)	Non-Obese (*n* = 168)	Obese (*n* = 103)	*p*-Value
Genotype			0.001
AA + AG	108 (64.3)	87 (84.5)	
GG	60 (35.7)	16 (15.5)	
Allele			0.001
A	146 (43.5)	119 (57.8)	
G	190 (56.5)	87 (42.2)	

Data are expressed as number (%). The *p*-values were calculated using the chi-squared test.

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
