# Peer review of "RANTES 59029A/G Polymorphisms Associated with Diabetic Compilations in Korean Patients with Type 2 Diabetes for over 15 Years"

_genes, 2021, doi:10.3390/genes12091445_

Round 1

Reviewer 1 Report

The authors propose that RANTES59029A is associated with diabetic complications in korean patients. They have compared the obese and non obese groups for the study.

1. The study does not have any control group. Without the control group it is difficult to compare it with the diabetes group. Since this is a retrospective study, they can include data for the control group.

2. The study shows that there is no significant difference in the blood parameters or the complications between the non obese and obese group. if this is the case why is the polymorphism important and what role does it have to play in the diabetic population.

3. The authors say there is a difference int he PCR product between the groups,but there is not image to see the difference. This should be added to the paper.

4. No data is available on the micro vascular complications. The complete data should be presented to compare the groups. The exact no of patients who had angina, myocardial infarction or CAD should be given.

Author Response

  1. The study does not have any control group. Without the control group it is difficult to compare it with the diabetes group. Since this is a retrospective study, they can include data for the control group.

à Thank you for your precise and insightful comment. We mostly agree your opinion. Because of the characteristics of patients in our clinic, we performed analysis in patients with T2D. We think it was also meaningful because of T2D was well-known risk factor for obesity. However, as reviewer’s opinion, it would be more valuable to compare with control group. We consider further evaluation in the future research. We have added these limitations in the revised manuscript.

[Revised] p. 10, lines 14–15 in the revised manuscript

The lack of control group of population without T2D is also a limitation of this study.

  1. The study shows that there is no significant difference in the blood parameters or the complications between the non obese and obese group. if this is the case why is the polymorphism important and what role does it have to play in the diabetic population.

à We appreciate the reviewer’s careful review. As the reviewer pointed out, there was no significant differences except body mass index between obese and non-obese groups. However, according to the presence of obesity there were significant differences in the prevalence of microvascular complications and genotypes of RANTES gene. Although obesity is a multifactorial disease, the results of this study suggested polymorphisms in the RANTES gene may influence the development of obesity and induce microvascular complications in patients with T2D.

  1. The authors say there is a difference in the PCR product between the groups, but there is not image to see the difference. This should be added to the paper.

à We thank the reviewer for this valuable comment. According the reviewer’s suggestion, we have added this contents as Figure 1 in the revised manuscript.

[Revised] p. 22, in the revised manuscript

Figure 1. Frequency of RANTES genotypes and alleles according to presence of obesity

  1. No data is available on the micro vascular complications. The complete data should be presented to compare the groups. The exact no of patients who had angina, myocardial infarction or CAD should be given.

à We appreciate the reviewer’s careful review. The prevalence of micro- and macro-vascular complications are presented in Table 2. In this study, the presence of CAD was defined in patients who had a history of stable angina, unstable angina, myocardial infarction, or intervention for coronary artery disease. The number of patients was small therefore, we did not conduct subgroup analysis according to the subtype of CAD. 

Reviewer 2 Report

Title of the article: "RANTES 59029A/G Polymorphisms are Associated with Diabetic Compilations in Korean Patients with Type 2 Diabetes for over 15 Years". The main focus is on patients from the Korean population. It is recommended that the introduction be rearranged in this direction. In this regard, it is necessary to cite well-known statistics on diabetes mellitus of the Korean population.

If the article is devoted to polymorphism in the RANTES 59029A/G gene, then the main emphasis in the introduction should be shifted from traditional risk factors to genetic risk factors for obesity and diabetes mellitus. Moreover, data on the effect of polymorphism of various genes (including RANTES) on the long-term duration of T2D in patients prone to complications should be presented.

In the introduction chapter, the authors did not give arguments for the need to study patients with diabetes mellitus with the AA or AG genotype of the RANTES gene. It is noteworthy that there are no links on the topic of the article (RANTES 59029A/G Polymorphisms are Associated with Diabetic Compilations in Korean Patients with Type 2 Diabetes for over 15 Years) for the last 5 years.

It is necessary to define the purpose of the study more precisely. As it seemed to me, this study is a continuation of previous studies. (A phrase from the article "Previous studies considering the effect of RANTES polymorphism on long-term duration of T2D in patients who are prone to complications have been insufficient"). It is correct to indicate this in the research goal. If this is not the case, then this duality should be removed in the introduction.

The methods presented are clear and reproducible, the references to the methods are correct.

The results of the study are not new in many ways and confirmed the previously known results of the study. The originality of the results is due to the fact that patients of the Korean population were studied.

No bias was found in the provision of results and conclusions.

The presented conclusions should be changed in accordance with the purpose of the study and explain the presence or absence of RANTES AA or AG genotypes in Korean patients with prolonged T2D for more than 15 years.

It is noteworthy that out of 25 articles, only 5 articles have been published in the last 5 years.

Author Response

Title of the article: "RANTES 59029A/G Polymorphisms are Associated with Diabetic Compilations in Korean Patients with Type 2 Diabetes for over 15 Years". The main focus is on patients from the Korean population. It is recommended that the introduction be rearranged in this direction. In this regard, it is necessary to cite well-known statistics on diabetes mellitus of the Korean population.

à We appreciate the reviewer’s careful assessment of our manuscript. According the reviewer’s suggestion, we have added these contents in the revised manuscript.

[Revised] p. 3 lines 5–7 and 1011 in the revised manuscript

An increase in the prevalence is observed in Korea as well. The prevalence of diabetes in Korea in 2017 was reported to be about 10% of the adult population, which was slightly higher than the global prevalence.

In Korea, diabetes was the sixth leading cause of death both in men and women.

[References for revision]

Oh, S.H.; Ku, H.; Park, K.S. Prevalence and socioeconomic burden of diabetes mellitus in South Korean adults: a population-based study using administrative data. BMC Public Health 2021, 21, 548, doi:10.1186/s12889-021-10450-3.

Jung, C.H.; Son, J.W.; Kang, S.; Kim, W.J.; Kim, H.S.; Kim, H.S.; Seo, M.; Shin, H.J.; Lee, S.S.; Jeong, S.J.; et al. Diabetes Fact Sheets in Korea, 2020: An Appraisal of Current Status. Diabetes Metab J 2021, 45, 1-10, doi:10.4093/dmj.2020.0254.

If the article is devoted to polymorphism in the RANTES 59029A/G gene, then the main emphasis in the introduction should be shifted from traditional risk factors to genetic risk factors for obesity and diabetes mellitus. Moreover, data on the effect of polymorphism of various genes (including RANTES) on the long-term duration of T2D in patients prone to complications should be presented.

à We thank the reviewer for this valuable comment. Obesity is known as a multifactorial disease. According to a previous study, genetic factors have been reported to account for about 40-70% of the causes of obesity. Previous studies have reported that polymorphisms in several genes are associated with obesity and diabetic complications in patients T2D. We have added these contents in the revised manuscript.

[Revised] p. 3 lines 1519 in the revised manuscript

Obesity is thought to be attributed to interactions between environmental and genetic factors. According to a previous report, genetic factors are responsible for approximately 40–70% of the etiology of obesity [5]. Previous studies have reported that polymorphisms in several genes are associated with obesity and diabetic complications in patients T2D [6-9].

[References for revision]

Yang, L.; Wu, L.; Fan, Y.; Ma, J. Vitamin D receptor gene polymorphisms in association with diabetic nephropathy: a systematic review and meta-analysis. BMC Med Genet 2017, 18, 95, doi:10.1186/s12881-017-0458-8.

Dong, J.; Ping, Y.; Wang, Y.; Zhang, Y. The roles of endothelial nitric oxide synthase gene polymorphisms in diabetes mellitus and its associated vascular complications: a systematic review and meta-analysis. Endocrine 2018, 62, 412-422, doi:10.1007/s12020-018-1683-4.

Zhuang, Y.; Niu, F.; Liu, D.; Sun, J.; Zhang, X.; Zhang, J.; Guo, S. Associations of TCF7L2 gene polymorphisms with the risk of diabetic nephropathy: A case-control study. Medicine (Baltimore) 2018, 97, e8388, doi:10.1097/MD.0000000000008388.

Montesanto, A.; Bonfigli, A.R.; Crocco, P.; Garagnani, P.; De Luca, M.; Boemi, M.; Marasco, E.; Pirazzini, C.; Giuliani, C.; Franceschi, C.; et al. Genes associated with Type 2 Diabetes and vascular complications. Aging (Albany NY) 2018, 10, 178-196, doi:10.18632/aging.101375.

In the introduction chapter, the authors did not give arguments for the need to study patients with diabetes mellitus with the AA or AG genotype of the RANTES gene. It is noteworthy that there are no links on the topic of the article (RANTES 59029A/G Polymorphisms are Associated with Diabetic Compilations in Korean Patients with Type 2 Diabetes for over 15 Years) for the last 5 years.

à Thank you for your precise and insightful comment. As we noted, the RANTES gene demonstrated an association with obesity and T2D incidence, respectively. However, studies on the effects of RANTES polymorphisms on patients with the long-term duration of T2D who are prone to complications have been lacking. Furthermore, studies evaluating the association among polymorphisms of RANTES gene, obesity, and T2D are also insufficient. We have added these contents in the revised manuscript.

[Revised] p. 4 lines 9–12 in the revised manuscript

Obesity and long-term duration of T2D are both risk factors for developing diabetes complications. However, studies on the effects of RANTES polymorphisms on patients with the long-term duration of T2D who are prone to complications have been lacking. Furthermore, studies evaluating the association among polymorphisms of RANTES gene, obesity, and T2D are also insufficient.

It is necessary to define the purpose of the study more precisely. As it seemed to me, this study is a continuation of previous studies. (A phrase from the article "Previous studies considering the effect of RANTES polymorphism on long-term duration of T2D in patients who are prone to complications have been insufficient"). It is correct to indicate this in the research goal. If this is not the case, then this duality should be removed in the introduction.

à We thank the reviewer for this valuable comment. In conjunction with previous comments of reviewer, we modified the last section of the introduction (purpose) in the revised manuscript.

[Revised] p. 4 lines 9–14 in the revised manuscript

However, studies on the effects of RANTES polymorphisms on patients with the long-term duration of T2D who are prone to complications have been lacking. Furthermore, there was no studies evaluating the association among polymorphisms of RANTES gene, obesity, and T2D. In the current study, we investigated whether RANTES polymorphism affected the prevalence of diabetic complications and its association with obesity in Korean patients that have had T2D for more than 15 years.

The methods presented are clear and reproducible, the references to the methods are correct.

The results of the study are not new in many ways and confirmed the previously known results of the study. The originality of the results is due to the fact that patients of the Korean population were studied.

No bias was found in the provision of results and conclusions.

à We appreciate the reviewer’s careful assessment of our manuscript.

The presented conclusions should be changed in accordance with the purpose of the study and explain the presence or absence of RANTES AA or AG genotypes in Korean patients with prolonged T2D for more than 15 years.

à According to reviewer’s comments, we have modified the conclusions of the revised manuscript.

[Revised] p. 10, lines 16–22 in the revised manuscript

In conclusion, the presence of RANTES AA or AG genotypes was associated with obesity in Korean patients with long-term T2D of more than 15 years. Furthermore, obesity in these patients demonstrated a significant correlation with the prevalence of diabetic microvascular complications. Our results support that the presence of AA or AG genotypes in RANTES gene may be involved as a risk factors for diabetic complications in Korean patients that have T2D for longer durations. A prospective study to confirm the present results and basic research to clarify the underlining mechanism are needed.

It is noteworthy that out of 25 articles, only 5 articles have been published in the last 5 years.

à We appreciate the reviewer’s careful assessment of our manuscript. Recent studies of RANTES polymorphisms in patient with T2D are lacking. However, we tried to search and have added recent articles during revision as reviewer’s comments.

Round 2

Reviewer 2 Report

The authors of the article supplemented the article with new data. All necessary corrections of the text of the article were made. The manuscript of the article has been significantly improved. The corrected version of the article can be published in Genes.